# China's Investment in the Nigerian Energy Sector: A Prognosis of the Dispute Settlement Paradigm

**Wen Xiang** [1,*] **and Olubayo Oluduro** [2]

1   Faculty of Law, University of Copenhagen, 1172 København, Denmark
2   Faculty of Law, Adekunle Ajasin University, Akungba-Akoko 342111, Nigeria; olubayo.oluduro@aaua.edu.ng
*   Correspondence: wen.xiang@jur.ku.dk

**Abstract:** Nigeria is one of the top countries of China's outward foreign direct investments in energy and power projects to meet the needs of China's fast-growing energy-intensive industries. Following several risks faced by investors to invest in countries with high levels of regulatory, judicial and political uncertainties that appeared in most African states, including Nigeria, contracting parties often take steps to advance and enhance their investment relations and investment climate through an agreement or bilateral investment treaties. This paper examines the China–Nigeria Bilateral Investment Treaty (BIT) and the investment arbitration framework in place in the energy sector. It includes a general analysis on China–African BITs and features common difficulties and possible ways of addressing them. It analyzes the adequacy or otherwise of these frameworks and the various protections afforded to the contracting parties or the host state and the investors. It contends that the current China–Nigeria BIT is lacking essential environmental and social aspects, including sustainable development, corporate social responsibility, transparency and respect for the human rights of host communities, for the promotion of better China–Nigeria investment relations. Notwithstanding the fact that there has not been any known energy dispute in China–Nigeria-related projects, this paper calls for the need for an effective and efficient dispute resolution mechanism to address future disputes between the parties, in order to promote a favorable investment climate for Chinese (and international) investors willing to invest in Nigeria. It advocates that the China–Nigeria BIT should be unambiguous and well drafted to cover issues that could best address investment disputes in the energy sector.

**Keywords:** China; Nigeria; BIT; investment; energy sector





## 1. Introduction

Nigeria is Africa's second-largest oil producer and the world's seventh-largest oil producer. Given the size of its oil and gas endowment in terms of proven reserves and its market position in the world oil and gas markets, Nigeria is one of the top five countries (Iran, Nigeria, Brazil, Kazakhstan and Australia) of China's Outward Foreign Direct Investments in energy and power projects aimed at meeting the needs of its fast-growing energy-intensive industries. As a result of China's growing appetite for oil to meet its rapid economic growth, it has to acquire oil from the oil-rich global communities. With the incursion of Chinese national oil companies (NOCs) into the energy investment in Africa, there is a need for bilateral agreements or treaties on both sides to take steps to promote and protect their interests. Following several risks faced by investors to invest in countries with high levels of regulatory, judicial and political uncertainties that appeared in most African states, including Nigeria, contracting parties often take steps to advance and enhance their investment relations and investment climate through an agreement or bilateral investment treaties. Disputes constitute a significant risk in the energy sector as it has a serious impact on investment, growth and development in the sector. The disputes usually arise from the investment statutes or an investment contract, and they may occur both when profits

are high and even when losses are high. Thus, it is important to consider how parties can manage these disputes whenever they occur to achieve better results.

This paper examines the China–Nigeria Bilateral Investment Treaty (BIT) and the investment arbitration framework in place in the energy sector. It analyzes the adequacy or otherwise of these frameworks and the various protections afforded to the contracting parties or the host state and the investors. It contends that the current China–Nigeria BIT is lacking essential environmental and social aspects, including sustainable development, corporate social responsibility, transparency and respect for the human rights of host communities, for the promotion of better China–Nigeria investment relations. Notwithstanding the fact that there has not been any known energy dispute in China–Nigeria-related projects, this paper calls for the need for an effective and efficient dispute resolution mechanism to address future disputes between the parties. It asserts that the chosen method of dispute resolution in the energy sector will go a long way in affecting the investment relationship of the contracting parties or the investors and the state. To reduce the cost, time and impact of disputes in the energy sector, this paper calls for the continued use of international arbitration as a means of settling disputes in order to promote a favorable investment climate for international investors willing to invest in Nigeria. It advocates that the China–Nigeria BIT should be unambiguous and well drafted to cover issues that could best address investment disputes in the energy sector.

## 2. China's Quest for Investment in Nigeria's Energy Sector

The rapid industrial growth of China in the past three decades—averaging nearly 12% per year—has resulted in the increasing demand for energy. In 2009, China edged out the United States to become the world's largest energy consumer (Pham 2011). China shifted from a net oil exporter in the early 1990s to a net oil importer in 1993 largely because Chinese NOCs' petroleum production could no longer meet the country's growing oil demand (Moreira 2013, p. 141). Also, China's foreign oil dependency reached 64.4 percent of its total demand in 2016 and is expected to rise to 76 percent by 2024 (Sun and Wang 2017; Patey 2017, p. 758; Lelyveld 2019). This is because of the widening gap between China's oil supply and demand and the projected gap between natural gas supply and demand. Being the world's fastest growing economy for more than three decades and more importantly based on the serious emphasis placed on industrial development (including manufacturing, mining, construction and utilities), which are largely energy-intensive, China has significantly increased its energy consumption (Pham 2011, pp. 2–3). While China's vast coal reserves will continue to provide most of its energy well into the foreseeable future, its oil and natural gas supply will, however, increasingly be unable to meet demand (Downs 2000, p. 6).

As a result of the rise in the domestic demand for energy due to its rapid economic growth, China's NOCs (majority-owned by the government but are not government-run) fully embraced the 'go out' or 'go global' policy, in which the Chinese government encourages its companies to invest and compete in international markets. Chinese NOCs search for, secure and promote long-term access to energy resources by investing in resource-rich countries around the world through joint ventures, service contracts, partnerships or strategic alliances with international oil companies (Pham 2011, p. 9). The China National Petroleum Corporation (CNPC), China Petroleum and Chemical Corporation (Sinopec) and China National Offshore Oil Corporation (CNOOC) are the three major Chinese NOCs that have continued to operate abroad particularly in global mergers and acquisitions in upstream oil and natural gas with strong support from the Chinese government to assist them in becoming national champions in the global economy.

Africa is a continent that is blessed with abundant natural resources and China needs to have access to Africa's energy resources, including its oil and gas for its internal industrial strategies and economic growth, to ensure a stable and prosperous nation (Dodo 2014, p. 757). Consequently, the Chinese government and its NOCs have continued to take serious steps to secure oil supplies from Africa. Currently, Africa provides about 22 percent

of China's total crude oil imports and serves as China's second-largest source of supply after the Middle East as of 2014 (Vasquez 2019). Nigeria has the largest economy in Africa and it is one of the most important countries of Chinese overseas oil investments. Nigeria is Africa's second-largest oil producer and the world's seventh-largest oil producer producing 2.4 million barrels per day, with one of the largest deposits of natural gas and oil, which provides more than 95% of Nigeria's earnings (Alao 2011; Nigerian National Petroleum Corporation 2018). It has over 35 billion barrels of oil, 187 trillion cubic feet of gas, 4 billion metric tons of coal and lignite as well as significant reserves of tar sands, hydropower and solar radiation, among others (Adenikinju 2008, p. 27). Given the size of its oil and gas endowment in terms of proven reserves and its market position in the world oil and gas markets, Nigeria is considered important by world energy agencies, fund managers and large corporate investors on account of its oil and gas reserves (Rapu et al. 2015, p. 25).

Even though Nigerian trade with China has grown substantially in the 21st century, its predominant focus is oil. In fact, around 87% of Nigeria's exports to China are oil and gas products (Egbula and Zheng 2011, p. 6). Chinese IOCs are currently investing in the downstream sector in Nigeria. In 1997, the CNPC began oil exploration in the Chad Basin under an agreement with the Nigerian National Petroleum Company (NNPC), and in 1998, the CNPC purchased two blocks—OML 64 and OML 66—in the Niger Delta region (Downs 2000, p. 22). In late 2004, it was pointed out that Sinopec signed two agreements, first with the NNPC to develop five exploration wells and second with the National Petroleum Development Corporation (NPDC) and the Nigerian Agip Oil Company (NAOC-Eni) to develop the Okono and Okpoho oil fields (Obi 2008, p. 422). In the same year, PetroChina signed an agreement with the NNPC for the daily supply of 30,000 barrels of oil to China for five years (ibid.). A big break, however, came in April 2006, when the China National Offshore Oil Corporation (CNOOC) acquired 'a 45 per cent stake in a Nigerian oil-for-gas field for US$2.27 billion and also purchased 35 per cent of an oil exploration license in the Niger Delta for US$60 million' (ibid.). This acquisition by the CNOOC in Akpo, a major oil field in Nigeria, was its largest foreign investment in the world until now and guaranteed the company 70 percent of the profits from OPL 246, with the NNPC taking 30 percent of the profits and 80 percent of the costs (ibid.). The CNOOC also agreed to spend USD 2 billion to build refineries and downstream infrastructure in Nigeria (Alao 2011, p. 22).

In continuation of its pursuits for oil abroad, the China National Petroleum Corporation (SINOPEC) took over Addax Petroleum in 2009 at a value of USD 7.2 billion (Tom-Jack 2016). In May 2010, the NNPC and the China State Construction Engineering Corporation (CSCEC) signed a Memorandum of Understanding (MoU) to jointly seek an estimated USD 23 billion in Contractor Financing and Supplier Credits from the China Export and Credit Insurance Corporation, SINOSURE and a consortium of Chinese Banks, for the construction of three refineries and a fuel complex. Under the terms, CSCEC agreed to cover 80% of the costs, with the NNPC financing the remaining 20% and the Lagos state government providing land and infrastructure. The refineries were expected to add some 750,000 barrels per day capacity to Nigeria's refining infrastructure and position the NNPC to engage profitably in the international trading of refined petroleum products (Europétrole 2010).

China and Nigeria are undoubtedly two regional giants. One of the key compliments China paid to Nigeria was designating Nigeria a 'strategic partner,' and Nigeria was the first African state to be designated as a strategic partner by China (Jackson 2019, p. 45). This was followed by the signing of a memorandum of understanding (MOU) on the establishment of a strategic partnership based on talks between Presidents Hu Jintao and Olusegun Obasanjo in the former's state visit to Nigeria in April 2006 (Jackson 2019, p. 45). Both countries have been well placed at each other's side on several international issues. In the recent talks of expanding China's biggest initiatives of the 21st century—the Belt and Road Initiative (BRI) elsewhere—the acting Chinese Consul-General in Lagos said that 'Nigeria, Africa's most populous nation, has a strategic role to play in the Belt and Road Initiative' (Xinhua News Agency 2019).

China's investments abroad in the energy sector, particularly in oil projects, are quite significant for several reasons. As noted by Downs, these investments are meant to fill the gap between domestic oil production and consumption; diversify China's import channels for energy security to reduce its dependence on the Middle East and its vulnerability to embargoes or blockades of Middle Eastern oil supplies due to the region's instability and volatility; gain greater control over China's foreign oil supplies; and insulate the Chinese economy from price hikes on the international market (Downs 2000, pp. 18–19). Such investments overseas are aimed at improving China's energy security by helping it to stabilize the economy in case of price fluctuations. In addition to the above, other factors that gave rise to China's quest for oil in Africa include the decline in China's imports from the Asia Pacific region, the latter having turned into a net oil importer; and the unique attributes of Africa's oil. Africa, as opposed to other parts of the world, boasts large untapped oil reserves, 'produces crude oil with low density and sulfur content, remains open to foreign oil investment, and continues to award production sharing agreements (PSAs) through which foreign oil companies can obtain equity oil (i.e., profit oil after cost recovery)' (Kong 2011). Chinese NOCs are also driven by the profit from the lucrative international oil business. Also important is that through its Belt and Road Initiative (BRI), China endeavors to export its developmental model with regard to a variety of policy areas (including taxation). Indeed, Chinese corporations, including Chinese state-owned enterprises, 'heavily shift their profits to tax havens, as well as to BRI jurisdictions [such as Africa] where, *inter alia*, statutory corporate tax rate are lower' (Segate 2022, p. 435). Interestingly, these same features also attract other oil-importing economies to African oil, particularly the Western oil majors, thereby making the continent's oil industry very competitive.

Suffice to say that the Western International Oil Companies (IOCs), such as Royal Dutch Shell, ExxonMobil, Chevron, Total and Eni, have long dominated Africa, dating back to the beginning of the 20th century, and so already captured attractive reserves and quality oil assets in the continent before the arrival of Chinese NOCs to Africa. As stated by two Chinese analysts, 'The energy companies in Europe and the United States have basically monopolized the oil industry in Nigeria ... which has squeezed the share of China's oil imports and caused more intense competition in energy projects than before' (Jackson 2019, p. 54). The Chinese NOCs had to contend themselves to searching for oil assets shunned by Western oil majors for their depleting reserves or high political risks, such as the Civil War or Western sanctions (Kong 2011). For example, the withdrawal of Western IOCs from Sudan as a result of the country's civil war, which appeared to make oil exploration very risky, and coupled with the U.S. sanctions against Sudan for human rights violations and for sponsoring terrorism, paved the way for the Chinese NOCs to step in to fill in the vacuum. The success recorded by the Chinese NOCs following the withdrawal of Western IOCs from Sudan in 1995 gave the NOCs the impetus to expand their operations in the continent. This aggressive push by the Chinese NOCs has made Africa an important continent for China's quest for investments in the energy sector.

## 3. Challenges for Nigeria–China Investment in the Energy Sector

The inconsistency in Nigerian government policies raises political and economic uncertainties. For example, former President Olusegun Obasanjo[1] once launched an initiative to secure investment from Chinese NOCs, to grant the latter oil exploration blocks in return for Beijing financing and building major infrastructure. Obasanjo tied Chinese financing for the project to the offer of four petroleum exploration blocks to state-owned CNOOC Ltd. The Chinese government and its NOCs signed onto the 'oil for infrastructure' projects, which consist of awarding oil contracts on favorable terms in exchange for the delivery of key infrastructure improvement projects. However, the new administration of President Umaru Yar'Adua who came to power following elections held in April 2007

---

[1] Olusegun Obasanjo was the President of Nigeria between 1999–2007.

cancelled and or suspended most of the oil-for-infrastructure contracts signed during the Obasanjo years, citing concerns about the lack of transparency (Egbula and Zheng 2011, p. 5; Fickling 2017). Thus, with new governments coming into power, contracts signed by former administrations with China were placed under review and, as the case may be, subsequently suspended or cancelled. President Yar' Adua changed the policy to 'oil for cash.'

The spate of militancy that has enveloped Nigeria is of serious concern. While the Boko Haram continues to strike in the northern part of Nigeria, the Militancy groups in the South, particularly in the Niger Delta region, will no doubt put off the would-be investors in the energy sector. For instance, the Movement for the Emancipation of the Niger Delta (MEND) threatened the Chinese oil companies against expansion into the Niger Delta by detonating a car bomb close to the Warri oil refinery, which coincided with the visit of the Chinese President Hu Jintao in April 2006 to Nigeria and the agreement of four oil-drilling licenses valued at USD 4 billion (Kong 2011). MEND stated that 'We wish to warn the Chinese government and its oil companies to steer well clear of the Niger Delta. The Chinese government by investing in stolen crude places its citizens in our line of fire.' (ibid.). In early 2007, MEND did not only attack oil installations but also kidnapped five Chinese oil workers for fourteen days. Also, fourteen Chinese workers were also kidnapped in the first two months of 2007 (Aboudou Kabassi 2012, p. 92). There were other attacks on several oil fields and the abductions of foreign expatriates, thus leading to the shutting down of operations by several oil MNCs and the reduction in the number of barrels that were produced in the Niger Delta region. Investment in the energy sector in Nigeria comes with a great risk of violence and instability capable of undermining development investment. This is similar to the situation in Ecuador, when in November 2006, local residents in Ecuador's Amazonian province of Sucumbios cut electricity to an oil field controlled by Andes Petroleum—owned by the CNPC and Sinopec—and took 40 Chinese oil workers hostage. In July 2007, another Indian-led protest occurred in Ecuador, against PetroOriental, owned by the CNPC and Sinopec where several people were injured (Moreira 2013, p. 151).

Given violent conflict in the Niger Delta region of Nigeria, it can be understandable why China has been acknowledging the need to support more military- and peace-building-related efforts in Nigeria in order to protect its investment and interests. China will continue to invest in this direction to be able to maximize economic investment in development (Toogood 2016). Arguably, China's concerns for Nigeria's security are partially self-motivated (Jackson 2019, p. 47). However, for China not to be accused of escalating and fueling the existing violence in the Niger Delta because of its current military efforts for Nigerian government, it has been suggested China should take steps to adopt and uphold conflict-sensitive business practices such as 'conducting conflict analysis, engaging in multi-stakeholder meetings in local communities, and building in grievance-redress mechanisms to assure that community grievances have a channel to reach key decision makers' (Toogood 2016). Such a conflict-sensitive policy will help to create confidence in the people in the oil-producing communities and remove the bias against China on its military support to Nigerian government, thus creating a social license for Chinese NOCs to operate peacefully in the Niger Delta region. In addition, China must be aware that if the Nigerian government breached some terms of the BIT in order to improve the situation of the Nigerian people, including its environment, it might find support in a line of arbitration cases. This was demonstrated in *Glamis Gold v. United States of America*.[2] In the words of Segate:

> in *Glamis Gold [v. United States of America* Award, IIC 380 (2009)], the ICSID tribunal rejected the claim of a Canadian company that the stringent regulations adopted at the federal and state levels on the conduct of mining operations in California would amount to indirect expropriation and breach of legitimate expectations of the foreign investor. The cultural value of the mining site as

---

2  Award, IIC 380 (2009).

ancestral land of a tribal community of Native Americans, together with compelling environmental considerations, was a factor in support of the legitimacy of the regulatory measures imposed by the United States' authorities in view of protecting the environment and landscape value of the relevant territory. (Segate 2021, p. 184)

No effective developmental projects, such as the laying of pipelines, could be carried out in the face of violent crimes. No doubt, the Chinese NOCs would be confronted with issues surrounding oil exploitation in the Niger Delta and the local resistance from the region, particularly 'the slippery and volatile politics of oil in the Niger Delta as the stakes for the control of access to the region's finite hydrocarbon resources climb higher' (Obi 2008, p. 430). In order to be able to actualize the BITs, there is the need for the Nigerian government to ensure better security in the country, particularly in the Niger Delta region. For the Chinese NOCs and other Western oil companies to be able to overcome these challenges, Bassey notes that they 'should not be agents of environmental despoliation and human rights abridgements,' and 'should avoid double-standards and not be a party to the multi-level corruption endemic in the industry' (ibid.). They must be ready to engage with the local people in the region, show respect and be sensitive to their plight and demands, by building trust and confidence between and with the communities (ibid.), and more importantly abide by the terms of the agreement entered into with the Nigerian government.

There is also a lack of transparency in relation to the opaque dealings in the oil sector leading to high levels of corruption in the industry. The United States' authorities are currently investigating China Petroleum and Chemical Corporation, known as Sinopec, over allegations that the state-controlled oil producer paid Nigerian officials about USD 100 million worth of bribes to resolve a USD 4 billion dispute between the Chinese oil company's Addax Petroleum unit in Geneva and the Nigerian government over drilling and other capital costs, tax breaks and a division of royalties between Addax and the Nigerian National Petroleum Corporation (NNPC) (Business & Maritime 2017). If corruption and transparency are not well handled, it may greatly hinder the mutual and long-lasting investment relationship between the two countries, hence the need for the inclusion of these provisions in the BIT.

It is noted that states 'forego needed environmental and social legislation that might negatively affect the value of foreign investment, rather than risk potential liability' (Gross 2003). Chinese NOCs are often accused of disregarding environmental protection in their investment and engaging in extracting natural resources without due consideration to the high environmental risks that are attendant to their exploratory activities, particularly in the ecologically fragile regions where they operate, thus further burdening the environment. Considering some of the negative views of China's oil ties in Africa, Dodo, citing Taylor, stated that 'China undermines good governance, environmental standard practices and human rights culture in Africa. And consequently, it props up corrupt and human rights violating regimes such as Zimbabwe, Sudan and Angola in the name of its so-called non-interference principle in other States' internal affairs' (Dodo 2014, p. 764). This situation appears worrisome since most African countries, including Nigeria, do not have the adequate legal framework in place to ensure environmental protection, and they neither have a strong judiciary to guarantee effective judicial remedies on environmental matters filed before it, nor administrative fines for non-compliance with the environmental standards obeyed. However, the adherents of the positive side of China's oil ties in Africa have discredited this view, stating that the opponents of the China-oil ties in Africa should focus their criticism more on the IOCs such as Shell, which enjoyed a monopoly in Nigeria's oil industry until its independence in 1960 and still remained the country's largest producer rather than China's oil companies who are latecomers and relatively small players in the industry (Downs 2007). Notwithstanding the views of the proponents of China's oil ties with Africa regarding the fact that some Western companies have behaved irresponsibly such as polluting the environment and engaging in corrupt practices in their host states, it

is argued that since oil exploration and production activities are known to be associated with a series of environmental problems with dire consequences on the host communities, such as the environmental damage of Ogoniland in the Niger Delta, Nigeria, as revealed in the UNEP Assessment of Ogoniland 2011, it is suggested that China should, in obedience to the BIT, ensure that it assists Nigeria by employing advanced technology in its current and future investments in the energy sector in order to combat environmental issues and promote development for the present and future generations.

## 4. Development and Problems of Nigeria–China BIT in the Energy Sector

Investment disputes between contracting parties are disputes between two different countries that arise on the basis of a bilateral investment treaty. It can be observed from Sino–African BITs that the settlement of investment disputes between African countries and China can be classified into three categories (Kidane 2016).

(1) Diplomatic or political settlement, which is the primary option for dispute resolution. As both contracting parties are represented, they are two independent subjects in international law, and diplomatic settlement has greater advantages and better reflects the contractual purpose of the BIT. International dispute resolution generally involves negotiation, consultation, investigation, mediation, conciliation, etc. There are no strict procedural requirements and the parties concerned do not have to adhere to formal requirements, which makes it easier to communicate and significantly reduces the cost and time required to resolve disputes. Most of the BITs in China and Africa provide for timely settlement through negotiation, and further specify the time limit for negotiation, and if no agreement can be reached within a certain period, the dispute can be resolved through subsequent arbitration by an ad hoc tribunal.[3]

(2) Mixed or ad hoc committees of representatives of both parties. This is usually proposed in a small number of China–African BITs as a dispute resolution body, which is a procedure between diplomatic consultation and referral to arbitration. If the dispute is not resolved within a reasonable period of time after the mixed committee has met and negotiated, it is referred to an arbitral tribunal for resolution. The process of meeting and the number of members of the mixed committee are not specified; only the representatives of the two parties are, which can be regarded as a kind of negotiating body with a lot of flexibility. The representatives of both sides may be composed of business negotiators and diplomats, which seems to be more conducive to the settlement of the agreement, but the mixed committee is, after all, another round of negotiation after the diplomatic negotiation, which is hardly effective in substance.[4]

(3) The arbitration. This is a common way of settling investment disputes between contracting parties. Arbitration is generally referred to as a quasi-judicial settlement because it has both contractual and judicial characteristics. The contractual feature is that before a dispute can be referred to arbitration, the parties must agree to submit the dispute to an arbitral tribunal; the judicial feature is that the parties agree to an arbitration clause that is legally binding.[5]

There are generally two options for arbitral tribunals in China–Africa BITs: ad hoc tribunals or international arbitration. Suffice to say that the composition of the arbitral tribunal, the establishment of the rules of arbitration procedure, the applicable law, the

---

[3]　Article 8 (2) of the Agreement between the Government of the People's Republic of China and the Government of the Federal Republic of Nigeria on the Reciprocal Promotion and Protection of Investments.

[4]　Article 9 (2) of the Agreement between the Government of the People's Republic of China and the Government of the Gabonese Republic on the Promotion and Reciprocal Protection of Investments, and Article 9 (2) of the Agreement between the Government of the People's Republic of China and the Government of the Kingdom of Morocco on the Encouragement and Reciprocal Protection of Investments.

[5]　Article 8 (2) of the Agreement between the Government of the People's Republic of China and the Government of the Republic of Kenya on the Encouragement and Reciprocal Protection of Investments, the provisions of Article 14 (2) of the Agreement between the Government of the People's Republic of China and the Government of the Republic of Mauritius on the Promotion and Reciprocal Protection of Investments, and the provisions of Article 9 (2) of the Agreement between the Government of the People's Republic of China and the Government of the Republic of South Africa on the Encouragement and Reciprocal Protection of Investments.

making and validity of the arbitral award and the sharing of costs are basically provided for in detail. In comparison, the arbitration mechanism for the settlement of disputes between the contracting parties is a better and more effective way for the contracting parties to freely safeguard their legitimate interests and fulfil their legal obligations under the investment agreement. Under the international investment, the arbitration of similar disputes in international investment can be based on the application of generally accepted rules in international law, thus maintaining a unified international investment environment.

Disputes in the energy sector usually arise from the BIT, investment statutes or an investment contract. Whitsitt and Bankes identified four generic types of disputes: '(1) disputes involving significant economic or political structural adjustment in the host state; (2) disputes triggered by the efforts of host state governments seeking to claim an enhanced share of resource rents; (3) disputes in which host state governments seek to enhance the environmental or social regulatory regime within which existing investments operate; and (4) disputes in which the host state government seeks to withdraw economic support mechanisms for a policy measure that was introduced to support a particular energy or environmental policy, such as a policy that seeks to reduce greenhouse gas emissions by favoring low carbon or alternative energy sources' (Whitsitt and Bankes 2013, p. 211). Claims may also be brought by investors on the grounds that a state has breached the fair and equitable treatment standard with regard to the investment or that the state expropriated the investment, via subsequent legislative or policy reform or somersault. Disputes may also arise as a result of the existence of political, environmental and security issues including issues regarding the non-payment of invoices and royalty fees; imposition of discriminatory legislation by the state, thereby destroying the value of an investor's property in the host state; delays, disruptions and cancellations (including force majeure claims); shareholder and joint venture disputes; disputes about the scope and transfer of rights; and issues about price as well as price adjustment claims in long-term supply contracts (Finizio 2016).

In view of the many risks faced by investors wanting to invest in countries with high levels of regulatory, judicial and political risk as obtained in most African states, foreign investors are often more concerned about the legal protection and security that are available to them during the period of their investments. Thus, BITs serve as valuable legal instruments to overcome these challenges. BITs have been defined by the United Nations Commission on Trade and Development (UNCTAD) as 'agreements between two countries for the reciprocal encouragement, promotion and protection of investments in each other's territories by companies based in either country' (UNCTAD 2004). Usually, a BIT is signed by two sovereign countries. Though an investor is not a party to the BIT, the BIT is concluded for the protection of investors from both countries, and an investor is entitled to claim that the host state violates the BIT, thereby giving rise to state responsibility (A Chinese Perspective: Africa—China Bilateral Investment Treaties 2015). A BIT creates a stable legal environment that fosters foreign direct investment and provides the investor with the opportunity to enforce its rights under the investment treaty against the host state through independent international investment arbitration (Bruce and Huard-Bourgois 2015). Nigeria and China could use BITs to advance and enhance their investment relations and investment climate. This is because BITs are designed to provide an important tool to reduce policy barriers limiting FDI, promote foreign investment, establish reciprocal rules for the treatment of firms and protection of investments (Peterson Institute for International Economics 2015, p. 5), and lay down terms of the relationship between host countries and the foreign investors in line with specific international standard norms. The Nigeria–China BIT 2001 is titled 'Agreement between the Government of the People's Republic of China and the Government of the Federal Republic of Nigeria for the Reciprocal Promotion and Protection of Investments.'[6] BITs are in practice meant to cover the specific areas of the definition of investment, scope of application, investment promotion and protection, and dispute settlement procedures (Oyeranti et al. 2010, p. 11), and more importantly sets

---

[6] Nigeria entered into BIT with China on 27 August 2001 which came into force on 18 February 2010.

out the commitments both countries expects each other to undertake and enforce as it concerns investment policies and the protection of investor rights. The Nigeria–China BIT is characterized by the following:

1.  Promotion of economic cooperation and encouragement of investors to make investments in their territories and admit such investments in accordance with its laws and regulations.
2.  Fair and equitable treatment.
3.  No expropriation against the investments of investors, unless (a) for the public interests; (b) under domestic legal procedure; (c) without discrimination; (d) against fair compensation.
4.  Guarantee to the investors the transfer of their investments and returns.
5.  Dispute to be settled through diplomatic channels and where the Parties cannot reach an agreement, the dispute shall be submitted to an arbitral tribunal constituted by the Contracting Parties.

Article 1 of the Nigeria–China BIT defines investment to mean every kind of asset invested by investors of one Contracting Party in accordance with the laws and regulations of the other Contracting Party in the territory of the latter, and in particular, though not exclusively, includes:

> (e) business concessions conferred by law or under contract permitted by law, including concessions to search for, cultivate, extract or exploit natural resources.

From the above, it can be argued that the energy sector falls within the definition of the protected type of investments under the said BIT. Nigeria–China BITs, like most investment treaties, provide for two separate dispute settlement mechanisms: one for disputes between the contracting states and the other for disputes between a host state and an aggrieved foreign investor (Idornigie 2016). The provisions on state–state arbitration regard the interpretation and application of the BIT, but as it concerns terms of dispute resolution arising from the investment, it is investor–state arbitration.[7] It is salutary that the BIT makes a provision for the arbitration of disputes between China and Nigeria outside of the courts in the host state. The dispute resolution process contained in the BIT provides for how interpretations are to be made on conflicts that arise with regard to issues such as coverage and scope, discriminatory practices and expropriation (Gentry and Ronk 2007, p. 44).

One of the major shortcomings of the Nigeria–China BIT is that it does not address more specific barriers necessary for the enhancement of the investment relationship in the energy sector, particularly environmental or social and sustainable development issues with the aim of balancing investor rights and public goods in a manner that is legitimate and transparent in order to ensure that the investment is consistent with sustainable development. Whilst the general provisions protecting investors' rights (coverage, expropriation, fund transfers, dispute resolution and others) can be of great benefit to all investors, including the energy sector, the adoption of energy specific investment agreements between China and Nigeria could serve as a platform for encouraging investment in the energy sector since many of the investment barriers in the energy sector can be most appropriately and directly addressed through sector-specific policies, rather than the more general provisions of the current BITs (Gentry and Ronk 2007, pp. 71–72).

The Nigeria–China BIT is also weak in terms of the protection offered to investors as it lacked commitments to labor, environmental and human rights standards, thus creating a substantial gap when compared with the 'hard' obligations contained in the recent Chinese BITs with the most developed countries, including the US. For instance, issues relating to the observance of human rights in the expropriation or exploitation of natural resources are not covered by the BIT. There is also no provision on corporate social responsibility (CSR) in the Nigeria–China BIT. Unfortunately, in Nigeria, there is an absence of statutory

---

7   China-Nigeria BIT, Arts 8 and 9.

provision on CSR other than provisions in corporate governance codes.[8] Thus, the BIT should contain provisions imposing duties on the investors to observe and perform CSR where they operate. Although the inclusion of explicit references to CSR in trade and investment agreements is a relatively recent phenomenon, some of the recent BITs in the field of investment, directly or indirectly through explicit reference made to existing CSR instruments such as the ILO MNE Declaration, the OECD Guidelines or the UN Global Compact, include the following:

1.  The Austria–Nigeria BIT signed in 2013 (not yet in force) expresses in the preamble the belief that responsible corporate behavior can contribute to mutual confidence between enterprises and host countries.
2.  The Netherlands–United Arab Emirates BIT signed in 2013 refers to the promotion of the OECD Guidelines.
3.  The Canada–Benin BIT signed in 2013 (not yet in force) mentions, in the core text, CSR as a guiding principle and commits 'each Contracting Party [to] encourage enterprises operating within its territory or subject to its jurisdiction to voluntarily incorporate internationally recognized standards' (Articles 4 and 16) (UNEP 2011, p. 25).
4.  The 2009 Norway model BIT (yet to be adopted) explicitly encourages investors to comply with the OECD Guidelines and the UN Global Compact (Peels et al. 2016).
5.  The Canada–Peru agreement that entered into force in 2009 made references to CSR in both the preamble and in several chapters of the body of the agreement, while also creating a forum to address CSR issues (UNEP 2011). For example, the investment section of the agreement, Article 810, provides that

> Each Party should encourage enterprises operating within its territory or subject to its jurisdiction to voluntarily incorporate internationally recognized standards of corporate social responsibility in their internal policies, such as statements of principle that have been endorsed or are supported by the Parties. These principles address issues such as labor, the environment, human rights, community relations and anti-corruption. The Parties therefore remind those enterprises of the importance of incorporating such corporate social responsibility standards in their internal policies. (UNEP 2011)

This provision is laudable as it linked the concept of CSR not only to labor, but also to the environment, human rights, community relations and anti-corruption. 'Given the serious environmental concerns in the many areas of Chinese investment in Africa, defining the environmental standards in investment treaties is not only critical but also feasible and politically expedient given China's increasing concern over environmental issues at home' (Kidane and Zhu 2014, p. 1072). Unfortunately, notwithstanding the serious environmental concerns inherent in several areas of China's investments in Nigeria, the Nigeria–China BIT has failed to define the environmental standards in the treaty. A look at the 2012 US BIT Model revealed an elaborate environmental provision when compared with the U.S. 2004 Model BIT, to show the value placed on environmental concerns. The Nigeria–China BIT must be crafted to protect and benefit the local deprived population where the projects are to be carried out. Kidnappings and demonstrations such as the ones in the Niger Delta and Ecuador underscore the importance of taking CSR/social accommodation seriously. Failure to do so has very grave implications, including damage to the reputation of the NOCs, kidnappings/killings of workers, delays and additional costs to the projects and loss of revenue (Moreira 2013, p. 152). Where every oil company in the Niger Delta practices good CSR, violent conflicts between the company and the host communities will cease and peace will reign supreme; oil production operations will continue to go on without any disruption; and the oil company will then be able to produce oil at its maximum production capacity, thereby accelerating Nigeria's oil exploitation to the climax (Kpolovie and Sado 2016, p. 43). Thus, there is a direct relationship between the intensity and scale of conflict

---

8　See section 172 of the English Companies Act 2006, and Idornigie (2016).

in the Niger Delta region and the CSR pattern of the oil MNCs operating in the region. Communities whose concerns are incorporated into CSR policies of the MNCs tend to have better cordial and harmonious relations with the operating companies, and conversely, corporate–community conflicts become more intense and enduring, sometimes leading to a complete breakdown of relations as a result of the failure of a corporation to be proactive in its CSR engagement with a host community (Aaron and Patrick 2013, p. 354).

Also lacking in the Nigeria–China BIT is the provision on corruption and transparency in their investment relationship. There is no gainsaying the fact that the oil industry in Nigeria is characterized by a lack of transparency and opaque dealings, and Chinese business culture is not better in this regard. For instance, the Nigerian National Petroleum Corporation (NNPC) had the poorest transparency record out of 44 national and international energy companies evaluated by the Transparency International (TI) and Revenue Watch Institute (RWI) based on research carried out in 2010. The NNPC was the only company to score zero (the average score was 65 percent) on organizational information disclosure, which included the provision details of deals agreed with governments and partners on energy projects (Brock 2011). The then Executive Secretary of the Nigeria Extractive Industry Transparency Initiative (NEITI), Mr. Waziri Adio, said that 90 per cent of the corrupt practices in Nigeria are being perpetrated in the oil and gas sector (Onwuemenyi 2016), and this accounted for the fact that despite the country's oil wealth, over 90 per cent of its citizens live on less than USD 2 per day. Permits, licensing, oil concession and contracts are awarded in return for bribe payments. The inclusion of transparency requirements in investment treaties, according to Kidane and Zhu, has the advantages of reminding and reinforcing the obligations on all sides, and perhaps more importantly, facilitating the enforcement of the obligations as part of the investor–state dispute settlement, such that where a host state fails to prosecute a public official who solicits a bribe from an investor, the investor could make use of the anti-corruption provisions in the BIT to seek redress for such violation (Kidane and Zhu 2014, p. 1079). It will also help to eliminate tension and conflicts in the Niger Delta region, combat resource curses, contribute to social and economic development and promote a transparent investment climate in the Nigeria's energy sector. Where the citizens in oil-producing countries are dissatisfied with governments and foreign investors perceived to be corrupt, this promotes political unrest and threatens oil supplies due to violent attacks, which hamper the MNCs to operate. As reported by Oil Revenue Transparency, in Nigeria, between 500 and 800,000 barrels of oil a day are lost due to attacks by militants' displeasure at the corruption of oil revenues and the secrecy of government budgets, made up mainly of oil revenues (Global Witness 2007, p. 4).

There is also the absence of the issue of sustainable development in the exploitation of natural resources in the Nigeria–China BIT. This is particularly important against the background of Chinese NOC large-scale investments in the energy sector where environmental issues, such as pollution, are a daily occurrence. The inclusion of such environmental requirements in the BIT is no doubt advantageous because it will help China improve its laws. It will also assist Nigeria to improve its environmental laws to protect the environment particularly as most of the energy projects affect large numbers of indigenous people of the Niger Delta, and will further help to set minimum standards by linking the environmental regulations to recognized human rights standards (Kidane and Zhu 2014, p. 1075).

In the words of Kidane, '[T]he reality is that today's China–Africa BIT regime is sporadic, outdated, uninformed by recent developments, incoherent, and even purposeless, China and African states need to consider renegotiating all three previous BITs as well as the latest generation ones. The negotiation must be informed by existing models, including the most recent Canadian BIT model for context, and, more importantly, the IISD Model BIT as it pertains to both the substantive and the dispute settlement provisions' (Kidane 2016, pp. 175–76). Thus, the Nigeria–China BIT ought to be reviewed to include international standards and best practices. For example, the two newly signed Canada–Cameroon and

Canada–Nigeria BITs are examples of 'new generation' BITs. They contain significant features[9], which include:

1. Promotion of 'sustainable development' in the preamble;
2. Protections to investments in pre- and post-establishment phases;
3. Expressly incorporating the international customary minimum standards of treatment of aliens (i.e., non-nationals);
4. Detailed guidance on the parameters of expropriation, preserving the regulatory space for certain public welfare measures;
5. Explicit recognition that states should not relax health, safety or environmental standards to attract investment;
6. Encouragement of corporate social responsibility;
7. Detailed procedural rules for investment arbitration, akin to the incorporation of institutional arbitration rules;
8. Three-year limitation periods and other procedural innovations.

The inclusion of these provisions in the revised BIT between Nigeria and China would be a great step forward toward a better investment dispute settlement and the promotion of the investors' relationship between them. For instance, the inclusion of the commitment to environmental standards in the BIT would help to reduce tension and conflicts in the volatile Niger Delta region where the exploitation of oil and gas takes place. In drafting investment treaties in the energy sector, it is expected that 'the risks and unknowns that attend sudden significant market fluctuations in the energy industry, technological advancements, sudden governmental intervention, or other catalysts for disputes' (Gaitis 2015, pp. 87–88) must be anticipated and taken into account. The drafters must also ensure that the draft treaties are perfectly unambiguous and well drafted such that the rights and duties of the parties are plainly set in stone regardless of future developments that are currently unknown (ibid., p. 88). For both countries to be able to gain significantly from their engagement in the energy sector, there is the need for them to look at ways to resolve their disputes in an amicable manner, to not strain the long-standing mutual relationship that had been existing between them. In this regard, the path for dispute resolution as contained in the BIT must be kept simple and workable.

## 5. Investment Arbitration in the Nigerian Energy Sector

Investor–state dispute settlement is traditionally a mechanism principally designed to provide the investor the opportunity to challenge the legality or otherwise of the actions of the host state or seek compensation for compensable breaches caused by the inadequacies in the domestic legal process of host state (Kidane 2016, p. 173). Settlement of investment disputes serves as the most important aspect of the international protection of investments. The traditional method for the settlements of disputes between states and foreign investors is the domestic courts followed by diplomatic protection after the exhaustion of local remedies (Schreuer 2013). The unsatisfactory characteristic of the traditional method has led to widespread acceptance of the arbitration between the host state and the investor with a major part of the investment arbitration taking place within the framework of the International Centre for Settlement of Investment Disputes (ICSID) (ibid.).

It is important to emphasize that China's long-standing stance of resolving bilateral disputes is first and foremost diplomatic and calls for mediation instead of litigation/arbitration. Thus, major Chinese state companies are basically attuned to diplomatic ways of settling disputes. Notwithstanding the fact that 'mandatory negotiation' is a legalistic standard that can be found in most BITs, China often strives to solve their disputes through bilateral discussion and direct negotiations between parties in accordance with historical facts and widely recognized international laws rather than submitting it to an arbitrator.

---

9 Agreement Between Canada and the Republic of Cameroon for the Promotion and Protection of Investments, 2014; Canada-Nigeria Foreign Investment Promotion and Protection Agreement (FIPA), 2014.

In order to create an investment-friendly environment, the Nigerian government has put in place measures, including laws and MOUs, to secure, promote and enhance foreign direct investment from China and other foreign investors. The most important legislation dealing with investment arbitration in Nigeria is the Nigerian Investment Promotion Commission (NIPC) Act.[10] This Act was promulgated in 1995 as the country's investment law and governs the entry of FDI into Nigeria. It allows for 100% foreign ownership in all sectors with the exception of oil and gas, where investment is limited to joint ventures and production-sharing agreements. Both local and foreign companies are to be registered with the Corporate Affairs Commission (CAC) following which they have to register with the NIPC. All investments with foreign participation are required to be registered with the NIPC to be covered by the treatment and protection clauses of the Act (Oyeranti et al. 2010, p. 14, See sections 17 and 27 of the NIPC Act). The Act specifically provides for the resolution of disputes arising between an investor and any government of the Nigerian Federation or any agencies of government in respect of an enterprise to which the Act applies. Section 26 (2) and (3)[11] provide that:

(2)   Any dispute between an investor and any Government of the Federation in respect of an enterprise to which this Act applies which is not amicably settled through mutual discussions may be submitted at the option of the aggrieved party to arbitration as follows:

    a.   In the case of a Nigerian investor, in accordance with the rules of procedure for arbitration as specified in the Arbitration and Conciliation Act; or

    b.   In the case of a foreign investor, within the framework of any bilateral or multilateral agreement on investment protection to which the Federal Government and the Country of which the investor is a national are parties; or

    c.   In accordance with any other national or international machinery for the settlement of investment disputes agreed on by the parties.

(3)   Where in respect of any dispute, there is disagreement between the investor and the Federal Government as to the method of dispute settlement to be adopted, the International Centre for Settlement of Investment Dispute Rules shall apply.

In addition to the above, there are other legislations in the energy sector that call for recourse to arbitration other than the court. The Petroleum Act 1969[12] in its regulation 41 provides that

If any question or dispute arises in connection with any licence or lease to which this schedule applies between the Minister and the Licensee or Lessee (including a question or dispute as to the payment of any fee, rent or royalty), the question or dispute shall be settled by Arbitration unless it relates to a matter expressly excluded from arbitration or expected to be at the discretion of the Minister.

Nigeria LNG (Fiscal Incentives, Guarantees and Assurances) Act.[13] Section 22 of the Act also states that

In the event of any dispute in respect of a substantial matter arising from the provision of this Act, the aggrieved Shareholder(s) in the Company shall issue a letter of notification to Government formally notifying Government and other shareholders of the dispute. The Government's representatives and one or more of the Company's Shareholders as the case may be, shall make serious efforts to resolve amicably such dispute. In the event of failure to reach amicable settlement

---

10   Cap N117, Laws of the Federation of Nigeria 2004.

11   In *Interocean Oil Development Company and Interocean Oil Exploration Company* v. *Federal Republic of Nigeria* (ICSID Case No. ARB/13/20), decision on Jurisdiction delivered on 29 October, 2014, the court held that statutory provisions like section 26 of the NIPC Act make standing offers to investors making a claim under the NIPC Act.

12   CAP P10 Laws of the Federation of Nigeria 2004.

13   CAP N87 Laws of the Federation of Nigeria 2004.

within 90 days of the date of the letter of notification mentioned above, such dispute may be submitted to arbitration before the International Centre for the Settlement of Investment Disputes.

Furthermore, Articles 8 and 9 of the China–Nigeria BIT provides, for the settlement of disputes between investors and the contracting party through the alternative dispute resolution (ADR), recourse to ad hoc arbitration, particularly if the dispute cannot be settled through negotiations. For sake of clarity, Article 9 provides the following:

1. Any dispute between an investor of the other contracting Party and the other Contracting Party in connection with an investment in the territory of the other Contracting Party shall, as far as possible, be settled amicably through negotiations between the parties to the dispute.
2. If the dispute cannot the settled through negotiations within six months, the either Party to the dispute shall be entitled to submit the dispute to the competent court to the Contracting Party accepting the investment.
3. If a dispute cannot be settled within six months after resort to negotiations as specified in Paragraph 1 of this Article it may be submitted at the request of either Party to an ad hoc arbitral tribunal. The provisions of this Paragraph shall not apply if the investor concerned has resorted to the procedure specified in Paragraph 2 of this Article.
4. Such an ad hoc arbitral tribunal shall be constituted for each individual case in the following way: each Party to the dispute shall appoint one arbitrator, and these two shall select a national of a third State which has diplomatic relations with the two Contracting Parties as the Chairman. The first two arbitrators shall be appointed within two months of the written notice for arbitration by either party to the dispute to the other, and the Chairman shall be selected within four months. If within the period specified above, the tribunal has not been constituted, either Party to the dispute may invite the Secretary General of the International Center for Settlement of Investment Disputes to make the necessary appointments.
5. The tribunal shall determine its own procedure. However, the tribunal may, in the course of determination of procedure, take as guidance the Arbitration Rules of International Center for Settlement of Investment Disputes.
6. ……….
7. The tribunal shall adjudicate in accordance with the law of the Contracting Party to the dispute accepting the investment including its rules on the conflict of laws, the provisions of this Agreement as well as the generally recognized principles of international law accepting by both Contracting Parties.

These statutes encourage the use of ADR mechanisms, including arbitration, as a means of promoting a favorable investment climate for international investors willing to invest in Nigeria. As shown above, the dispute settlement mechanism in the Nigeria–China BIT covered 'any dispute…in connection with an investment.'[14] The BIT provided a mandatory negotiation step before arbitration. It also contains provision for the investor to choose from either domestic proceeding or international arbitration if the dispute cannot be settled through negotiations.[15] Once the investor has submitted the dispute to the competent court of the host state, it can no longer bring a claim to an ad hoc international arbitration tribunal. Thus, an investor has to choose one way or another at the cross-road (termed 'fork-in-the-road' provision) for ways of remedying in the event that it has suffered a loss from a host state's action or inaction, which may amount to a violation of the provisions of the BIT (A Chinese Perspective: Africa—China Bilateral Investment Treaties 2015). With the arbitration clauses contained in these statutes, contracting parties are bound by such clauses in case of disputes. In *Onward Enterprises Ltd.* v. *M.V Matrix*,[16] the Nigerian Court of Appeal stated that 'Once an arbitration clause is retained in a contract

---

14　China—Nigeria BIT, Art. 9(1).
15　China—Nigeria BIT, Art. 9(2) & (3).
16　Case of *Onward Enterprises Ltd. V M.V Matrix* [2010] 2 NWLR (Part 1179) at 530, 558.

which is valid and the dispute is within the contemplation of the clause, the court should give regard to the contract by enforcing the arbitration clause. It is therefore the general policy of the court to hold parties to the bargain which they had entered.'

Until now, there has not been any known energy dispute in Nigeria–China-related projects. Also, no publicly available awards have been issued against Nigeria under any of the BITs. Nigerian courts are yet to be called upon to enforce an investment treaty award against Nigeria (Ufot 2013). So far, three cases have been filed against Nigeria at ICSID by foreign investors in respect of its investment treaties. However, two were discontinued before the conclusion of the arbitration proceedings: *Guadalupe Gas Products Corporation v. Nigeria*[17] and *Shell Nigeria Ultra Deep Limited v. Federal Republic of Nigeria*.[18] However, in *Interocean Oil Development Company and Interocean Oil Exploration Company v. Federal Republic of Nigeria*[19], the dispute was brought by two American oil companies, which are accusing the country's state-owned oil company of illegally seizing control of a separate Nigerian firm through which they held a prospecting lease. On 6 October 2020, an ICSID tribunal considered indirect expropriation and breach of customary international law claims brought by the Interocean Oil Development Company and Interocean Oil Exploration Company (claimants), against Nigeria, pursuant to the Nigerian Investment Promotion Commission Act (NIPCA). The claimants maintained that the collective actions of the respondent's agencies and domestic courts, as well as the private actions of Festus Fadeyi, which allegedly led to the expropriation of their investments in Nigeria, were attributable to the respondent. The tribunal in its award dismissed the claims and awarded costs in the sum of USD 660,129.87, in favor of the respondent. The Tribunal similarly dismissed all jurisdictional objections raised by the respondent on the grounds that non-registration is not a sufficient reason to deny the ICSID Tribunal the jurisdiction to adjudicate over the dispute.

Following the decline by the Chairman of the Administrative counsel of the International Centre for Settlement of Investment Disputes of the proposal for disqualification of the three members of the Tribunal on 27 October 2017, the proceeding has resumed pursuant to ICSID Arbitration Rule 9(6). Notwithstanding the above, Nigerian courts have continued to show support with regard to the enforcement of arbitration agreements and arbitral awards.[20]

In addition to the pro-enforcement bias with regard to commercial arbitration by Nigerian courts, there are other legal frameworks in place for arbitration in Nigeria. There is the Arbitration and Mediation Act,[21] modeled on the United Nations Commission on International Trade Law (UNCITRAL) Model Law, and it applies throughout the Nigerian Federation. Nigeria has also ratified the International Centre for Settlement of Investment Disputes (ICSID) Convention on 23 August 1965 and the convention was re-enacted as a local legislation in Nigeria through the International Centre for Settlement of Investment Disputes (Enforcement of Awards) Act.[22] As of November 2022, ICSID has 158 contracting states, including China and Nigeria, and has administered roughly 70% of all known investor–state cases (Kinnear 2023, p. 35). The ICSID Act provides for the enforcement of ICSID awards directly at the Nigerian Supreme Court as the court of first instance by the party seeking its recognition and enforcement. Arguably, this makes it the fastest procedure for enforcing an arbitral award in Nigeria as there is little or no room for objections to the enforcement of the award (Okoronkwo 2023) or review by domestic courts.

---

[17]   ICSID Case ARB/78/1 – discontinued on 22 July 1980.

[18]   ICSID Case ARB/07/18 – discontinued on 1 August 2011.

[19]   ICSID Case No. ARB/13/20, registered on 9 September 2013. See (Echebima 2020), at ICSID tribunal dismisses claims of Interocean Oil Development Company and Interocean Oil Exploration Company against Nigeria while upholding its jurisdiction to hear the claims solely based on Nigeria's domestic investment statute – Investment Treaty News (iisd.org).

[20]   *Continental Sale Limited v R Shipping Inc* (2013) 4 NWLR (part 1343), p. 67.

[21]   Arbitration and Mediation Act 2023 which repealed the Arbitration and Conciliation Act Cap A18, Laws of the Federation of Nigeria, 2004.

[22]   Cap I 20, Laws of the Federation of Nigeria, 2004.

Furthermore, Nigeria became a signatory to the New York Convention on the Recognition and Enforcement of Foreign Arbitral Award (the New York Convention) 1958 on 17 March 1970, and the convention came into force in June 1970. As of January 2023, the convention has 172 state parties,[23] including China and about 33 African countries of which Nigeria is a signatory. The New York Convention applies in Nigeria by virtue of Section 60 of the Arbitration and Mediation Act 2023. Section 60 of the Act provides that

> Without prejudice to sections 57 and 58 of this Act, where the recognition and enforcement of any award made in an arbitration in a Country other than Nigeria is sort, the New York Convention on the Recognition and Enforcement of Foreign Awards set out in the second schedule to this Act applies to an award, provided that the-
>
> (a)   country is a party to the New York Convention;
> (b)   and differences arise out of a legal relationship, whether contractual or not, considered commercial under the laws of Nigeria.

Nigeria has made a reciprocity reservation. Thus, only awards made in contracting states that attempt to recognize and enforce awards made in other contracting states, including Nigeria, will be recognized and enforced in Nigeria (Okoronkwo 2023). The fact that both countries are parties to the ICSID Convention and the New York Convention provide strong assurances to the investors that arbitral awards will ultimately be enforced against the host state (Ofodile 2013). This is in addition to the fact that disputing parties are allowed to choose their arbitrator and are granted increased control over the arbitral process. This is unlike where the dispute is brought before domestic courts of the host state for settlement.

In addition, Nigeria has viable arbitration institutions in place for the successful conduct of arbitration. These include the Regional Centre for International Commercial Arbitration, Lagos; the Chartered Institute of Arbitrators, Nigeria Branch; the Society for Construction Industry Arbitration; the Maritime Arbitrators Association of Nigeria; the Arbitration Commission of the ICC Nigerian National Committee; and the Lagos Court of Arbitration, among others.

With the level of investment of the Chinese government in the energy sector in Nigeria, and given the size and complexity, the high-risk and myriad commercial and technical arrangements involved in most energy sector projects, and the fact that many energy contracts cover a long period of time, such as LNG contracts that usually have 20-year terms (Sun and Wang 2017), it is inevitable that a variety of energy-related issues and disputes will continue to emerge. Having signed a BIT with China with the aim of offering full protection for the investors, and coupled with the arbitration framework and national infrastructure and arbitration institutions in place, investment arbitration as a legitimate dispute resolution mechanism needs to be promoted and encouraged in line with the provisions of the BIT agreement for the purpose of settling future or potential disputes that may arise from increased investment in the energy sector between Nigeria and China. This is because this mechanism is capable of promoting a continuing better, friendly and cooperative investment relationship between the parties, particularly if the parties wish to continue with their commercial relationship.

Notwithstanding the importance of investment treaties generally, there is an increasing opposition to it with some African countries now taking steps to withdraw from treaty obligations including by cancelling BITs. For instance, South Africa in October 2012 cancelled its BITs with Belgium–Luxembourg, Spain, Germany, Switzerland, the Netherlands and Denmark (Finizio 2016). As an alternative to BIT protections, South Africa published a draft Promotion and Protection of Investment Bill, which was tabled in parliament in July 2015 and finally enacted into law in 2015 titled 'Protection of Investment Act' Act

---

23    New York Convention 1958 Guide, *New State Party to the New York Convention: Timor-Leste*, at New State Party to the New York Convention . . .—News—New York Convention Guide 1958 (newyorkconvention1958.org).

22 of 2015.[24] According to the Government of South Africa, this step has been taken in order to 'update and modernise South Africa's legal framework to protect investment in South Africa,' to clarify the 'strong protection' already provided to foreign investors by South Africa's national legislation and to better balance 'the rights of investors to protection with the right of Government to regulate and safeguard the public interest.'[25] It is also aimed at protecting foreign investments by way of municipal legislation rather than by way of international treaty. Also, Egypt amended its Investment Law No. 8/1997 in 2015, by removing reference to investor–state treaty arbitration, among others (Finizio 2016). Ecuador has not only denounced most of their BITs and withdrew from the ICSID, but also set up citizens' commissions to audit the BITs, with the finding that these have only benefited investors (Bantekas 2021, pp. 357–58). Also, some Latin American countries such as Bolivia and Venezuela have terminated BITs or withdrawn from the ICSID. Notwithstanding these developments, Nigeria is certainly not yet ripe for this, due to the weakness and unnecessary delay inherent in its judicial system.

## 6. Conclusions

The China–African BITs needs better improvements in its dispute settlement provisions. Firstly, if the BITs in China and Africa do not provide for a means of settlement other than diplomatic consultation, a special arbitral tribunal or international arbitration should be added to the list of means of settlement. For example, the rules of the International Centre for Settlement of Investment Disputes (ICSID) or the International Commercial Arbitration Rules should be applied by reference. At the same time, the scope of application of the law on arbitration should be further clarified and expanded, to facilitate the timely and effective resolution of disputes between the parties. Bilateral investment agreements are applicable as a matter of course in the settlement of disputes. In addition, international treaties to which both are parties and the generally recognized principles and rules of the International Court of Justice jointly entered into or approved by both parties on the basis of their free and equal will, and are legally binding on both parties, to the same extent, should also be applicable (Guang 2009).

Secondly, the prerequisite of 'exhaustion of local administrative or judicial remedies' should be removed as an optional clause. The scope of disputes in which local administrative review procedures are exhausted can be relatively narrowed. At present, the exhaustion of the local remedies rule in China–Africa BITs should be treated in a more flexible manner, as the legal systems of most African countries are not very transparent. Requiring investors to exhaust local remedies rules makes it difficult for investors to understand and familiarize themselves with the relevant substantive and procedural laws of the host country. In addition, some African countries have an unstable legal environment, which poses greater legal risks. It is more reasonable to make the exhaustion of local remedies an optional condition for the exclusion of arbitration, rather than a mandatory pre-condition. Furthermore, the scope of disputes that can be resolved by arbitration should no longer be limited to expropriation disputes. Other disputes beyond the amount of compensation for expropriation may also be subject to arbitration, while the attitude of accepting international arbitration should be relaxed, as international arbitration has its unique advantages in resolving international investment disputes. To a large extent, international investment arbitration avoids the politicization of investment dispute resolution, depriving investment law of its historical justification for the use of forceful claims (Segate 2021).

The issue of the validity of the legal basis of arbitral awards is clarified through the interpretation of Central African BITs or arbitration agreements, and the procedural aspects of the application of these laws are further elaborated. The question of the validity of the applicable legal basis for arbitration cannot be supplemented or clarified when settling

---

[24] The Protection of Investment Act 22 of 2015.
[25] 'Minister Rob Davies on Promotion and Protection of Investment Bill', South African Government Online (28 July 2015), quoted in Woolfrey (2016, pp. 277–78).

investment disputes between investors and host countries through ad hoc arbitral tribunals, as the rules and procedures applicable to arbitration are not clearly set out in many of the preceding articles of the Central African BITs. However, this problem can be avoided to a certain extent because international investment tribunals have clear arbitration rules, e.g., Article 42 of the Convention on the Settlement of Investment Disputes between States and Nationals of Other States specifies the applicable legal rules for arbitration, and the ICSID in its arbitration practice, the domestic law of a contracting state, investment treaties and the rules of general international law apply equally (Guang 2011).

BIT protections and recourse to international arbitration toward protecting contracting parties/investors rights remain germane regarding China's investment in the Nigeria's energy sector. Considering the fact that Chinese investors are usually hesitant to litigate in African courts due to the prevailing perceptions among them of corruption, delay, inconsistency or vulnerability to bias in African courts; the difficulty in enforcing judgments made by African courts (sometimes unjustifiably); and their lack of knowledge about litigation processes in Africa due to the diversified legal systems in the region, when a dispute arises, the Chinese investors either give up the investment or seek to settle the dispute through unjustifiable means, such as bribing local officials (Zhu 2013). These fears and worries also hold on the part of the host states. While working on building strong domestic institutions such as the judiciary (court congestion leading to delay in trials, corruption, inefficiency and lack of expertise to deal with disputes arising from complex and transnational business transactions), we therefore call for the continued use of international arbitration as a means of settling disputes in order to promote a favorable investment climate for international investors willing to invest in Nigeria's energy sector. Also, while this paper condemned the weak regime of human rights in the BITs between China and Nigeria, it is submitted that irrespective of BITs, states should be implementing their human rights obligations, and no other treaty obligation should be allowed to curtail, directly or indirectly, human rights (Bantekas 2021). Hence, BITs should be interpreted in a manner that is consistent with customary human rights obligations (ibid.). Furthermore, environmental clauses are necessary in BITs to 'ensure that investment instruments do not impede a State's "right to regulate" the environment, as well as to prevent the State from failing to enforce its environmental regulations in order to attract new investment' (Gentry and Ronk 2007, p. 45). Indeed, environmental governance in Nigeria is very weak and inadequate to protect against potential environmental damage, hence the need to have incorporated in the Nigeria–China BIT a strong environmental clause. Prolonged, unresolved disputes in the energy sector can paralyze development and restrict investment in infrastructure (Bruce and Macmillan 2004, p. 2) to the detriment of countries such as Nigeria that have historically experienced a lack of investment and growth in their economy. A lack of an effective and efficient dispute resolution where players in the industry can constructively resolve their disputes can be destructive to the energy sector.

**Author Contributions:** Conceptualization, W.X. and O.O.; methodology, W.X. and O.O.; formal analysis, W.X. and O.O.; investigation, O.O.; resources, W.X. and O.O.; writing—original draft preparation, W.X. and O.O.; writing—review and editing, W.X. and O.O.; project administration, W.X. and O.O.; funding acquisition, W.X. All authors have read and agreed to the published version of the manuscript.

**Funding:** This research was funded by Sino-Danish Center for Education and Research.

**Data Availability Statement:** Not applicable.

**Conflicts of Interest:** The authors declare no conflict of interest. The funders had no role in the design of the study; in the collection, analyses, or interpretation of data; in the writing of the manuscript; or in the decision to publish the results.

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
