# Peer review of "China’s Investment in the Nigerian Energy Sector: A Prognosis of the Dispute Settlement Paradigm"

_laws, 2023_

Round 1

Reviewer 1 Report (Previous Reviewer 1)

The author(s) have fully complied with my suggestions, my assessment of the contribution is positive. It is a fine piece of research both from methodology and contents-wise, with interesting elements of novelty and originality.

No comments regarding quality of English language.

Author Response

Dear Reviewer, 
Thank you so much for your kind comments. We have made final check on our manuscript and made changes on editing of grammar, wording as well as references. 
We are so grateful for your time on our paper, many thanks! 
Best regards,

Reviewer 2 Report (Previous Reviewer 2)

Improved compared to the original version.

Minor grammar issues, which however do not hinder the reading.

Author Response

Dear Reviewer, 
Thank you so much for your kind comments. We have made final check on our manuscript and made changes on some parts of the body of the paper, editing of grammar, wording as well as references. All changes are marked in the manuscript for your kind review. 
We are so grateful for your time on our paper, many thanks! 
Best regards,
Authors

This manuscript is a resubmission of an earlier submission. The following is a list of the peer review reports and author responses from that submission.

Round 1

Reviewer 1 Report

The contribution address a topic of particular interest which has not been thoroughly analysed in the literature.

That being said, the work is not ripe for publication in its current format. The legal analysis is reduced to a minimum and revolves around considerations - such as the absence of CSR rules in the 2001 China-Nigeria BIT - which are obvious from a development of investment law perspective. Given the lack of arbitral case law against Nigeria, the Author should have at least compared the China-Nigeria BIT with other Chinese and Nigerian BITs, so as to highlight the peculiarities of the investment-law relationship between the two countries and, possibly, how such peculiarities impact on the energy sector.

Moreover, the literature the author refers to is limited to regional-specific works, without any reference to general investment law literature. A thorough study of authors like Schreuer, Vinuales, McLachlan, Tanzi or Dolzer is necessary to understand the criticalities that a BIT may have and, thus, provide a good piece of literature on the matter.

Reviewer 2 Report

·       Key literature is missing, e.g. https://ww3.lawschool.cornell.edu/research/ILJ/upload/Kidane-final.pdf

·       No emphasis on “the Chinese way” of solving bilateral disputes, which is first and foremost diplomatic and calls for mediation instead of litigation/arbitration, all the more so as major state companies are basically akin to diplomatic arms for the Chinese government. “Mandatory negotiation” is a legalistic standard that can be found in most BITs, but my point is that even beyond mandatory negotiation, China will try to solve the matter escalating it to higher spheres of politics, rather than submitting it to an arbitrator.

·       The author does mention that China seeks a stable environment for its investments and in order to secure such stability, tends to disregard societal problems and to contribute to private militarization of Nigerian territories. However, the author fails to fully explain why. One reason is that China must be aware that if the Nigerian government breached some terms of the BIT in order to improve the situation of the Nigerian people, including environmentally, it might find support in a line of arbitration cases. See e.g. https://repository.uchastings.edu/hastings_environmental_law_journal/vol27/iss1/5/ p. 184. To put it simple, it wouldn’t be indirect expropriation.

·       English inaccuracies and mistakes, even starting from the abstract (e.g. “these framework”) but also all throughout the text (e.g. “importer in 1993 and is foreign oil”). Also, at times the language is too informal (e.g. “China has a lot of interests in Nigeria”)

·       Several factual inaccuracies. For instance, the author claims that Nigeria is Africa’s largest oil producer (p 3), while in fact Angola has overtaken Nigeria in early 2022.

·       Some contextual references are missing to China’s broader aims throughout the BRI, for example about the fact that through the BRI, China endeavours to export its developmental model (see e.g. https://library2.um.edu.mo/etheses/991010238079006306_ft.pdf  p 435) with regards to a variety of policy areas (including taxation), but indeed not necessarily paying due attention to its model not being always “exportable” – particularly from a developmental complexity, and if sustainability is to be valued.

·       Some comparative flavours are also warranted. Consider e.g. https://journals.sagepub.com/doi/10.1177/186810261304200106

·       The same goes with deeper diplomatic background. Check e.g. https://www.airuniversity.af.edu/Portals/10/JEMEAA/Journals/Volume-01_Issue-2/JEMEAA_01_2_Jackson.pdf

·       Some references are pretty outdated. For instance, the author claims that “Nigerian courts are yet to be called upon to enforce an investment treaty award 589 against Nigeria”, but the cited blog post traces back to 2013

·       At p. 17, the author mentions that “as an alternative to BIT protections, South Africa published a draft Promotion and Protection of Investment Bill which was tabled in parliament in July 2015 and has yet to be enacted. If passed into law, foreign investors will have to be submitting their disputes before national courts or relevant authorities.” This is actually done in view of an upcoming major revolution in state approaches to corporate human-rights violations: the binding treaty on business and human rights. Refer extensively to https://lawecommons.luc.edu/cgi/viewcontent.cgi?article=1237&context=lucilr (e.g. p. 61). Same within the EU: https://journals.sagepub.com/doi/10.1177/1023263X19896917 (e.g. pp 83-84)

·       The Conclusion should be strengthened considerably. The author simply states that investment arbitration should continue to be regarded as a privileged forum for addressing (energy) disputes between China and Nigeria, but I expected a resolute concluding thought on what kind of amendments would be necessary to both the BIT itself and the parties’ behavior/expectations for this ADR system to actually work. True that there are sound reasons to mistrust African courts, but there are equally sound reasons why the parties are reluctant to make the most of arbitration, or at least to solve the underlying social-balancing issues (environment, human rights, poverty, etc.) through it.